# COVID-19 Vaccine Uptake and Effectiveness by Time since Vaccination in the Western Cape Province, South Africa: An Observational Cohort Study during 2020–2022

**DOI:** 10.3390/vaccines12060628

**Published:** 2024-06-05

**Authors:** Reshma Kassanjee, Mary-Ann Davies, Alexa Heekes, Hassan Mahomed, Anthony J. Hawkridge, Erna Morden, Theuns Jacobs, Cheryl Cohen, Harry Moultrie, Richard J. Lessells, Nicolette Van Der Walt, Juanita O. Arendse, Nicole Wolter, Sibongile Walaza, Waasila Jassat, Anne von Gottberg, Patrick L. Hannan, Daniel R. Feikin, Keith Cloete, Andrew Boulle

**Affiliations:** 1Centre for Infectious Disease Epidemiology and Research, School of Public Health, Faculty of Health Sciences, University of Cape Town, Cape Town 7925, South Africa; mary-ann.davies@uct.ac.za (M.-A.D.); alexa.heekes@westerncape.gov.za (A.H.); andrew.boulle@uct.ac.za (A.B.); 2Division of Public Health Medicine, School of Public Health, Faculty of Health Sciences, University of Cape Town, Cape Town 7925, South Africa; 3Centre for Infectious Diseases Research in Africa, Faculty of Health Sciences, University of Cape Town, Cape Town 7925, South Africa; 4Health Intelligence, Western Cape Department of Health and Wellness, Cape Town 8000, South Africa; erna.morden@westerncape.gov.za (E.M.); theuns.jacobs@westerncape.gov.za (T.J.); 5Division of Health Systems and Public Health, Department of Global Health, Faculty of Medicine and Health Sciences, Stellenbosch University, Tygerberg 7505, South Africa; hassan.mahomed@westerncape.gov.za (H.M.); juanita.arendse@westerncape.gov.za (J.O.A.); 6Metro Health Services, Western Cape Department of Health and Wellness, Cape Town 8000, South Africa; 7Rural Health Services, Western Cape Department of Health and Wellness, Cape Town 8000, South Africa; anthony.hawkridge@westerncape.gov.za; 8School of Public Health, Faculty of Health Sciences, University of Cape Town, Cape Town 7925, South Africa; 9Centre for Respiratory Diseases and Meningitis, National Institute for Communicable Diseases of the National Health Laboratory Service, Johannesburg 2192, South Africa; cherylc@nicd.ac.za (C.C.); nicolew@nicd.ac.za (N.W.); sibongilew@nicd.ac.za (S.W.); annev@nicd.ac.za (A.v.G.); 10School of Public Health, Faculty of Health Sciences, University of the Witwatersrand, Johannesburg 2193, South Africa; 11Centre for Tuberculosis, National Institute for Communicable Diseases of the National Health Laboratory Service, Johannesburg 2192, South Africa; harrym@nicd.ac.za; 12KwaZulu-Natal Research Innovation & Sequencing Platform, University of KwaZulu-Natal, Durban 4001, South Africa; lessellsr@ukzn.ac.za; 13Emergency & Clinical Services Support, Western Cape Department of Health and Wellness, Cape Town 8000, South Africa; nicolette.vanderwalt@westerncape.gov.za; 14School of Pathology, Faculty of Health Sciences, University of the Witwatersrand, Johannesburg 2193, South Africa; 15Health Practice, Genesis Analytics, Johannesburg 2196, South Africa; waasilaj@genesis-analytics.com; 16Division of Public Health Surveillance and Response, National Institute for Communicable Diseases of the National Health Laboratory Service, Johannesburg 2192, South Africa; 17Division of Epidemiology and Biostatistics, School of Public Health, Faculty of Health Sciences, University of Cape Town, Cape Town 7925, South Africa; hnnpat002@myuct.ac.za; 18Department of Immunizations, Vaccines, and Biologicals, World Health Organization, CH-1211 Geneva, Switzerland; feikind@who.int; 19Western Cape Department of Health and Wellness, Cape Town 8000, South Africa; keith.cloete@westerncape.gov.za

**Keywords:** COVID-19, SARS-CoV-2, vaccine effectiveness, South Africa, cohort, observational

## Abstract

There are few data on the real-world effectiveness of COVID-19 vaccines and boosting in Africa, which experienced widespread SARS-CoV-2 infection before vaccine availability. We assessed the association between vaccination and severe COVID-19 in the Western Cape, South Africa, in an observational cohort study of >2 million adults during 2020–2022. We described SARS-CoV-2 testing, COVID-19 outcomes, and vaccine uptake over time. We used multivariable cox models to estimate the association of BNT162b2 and Ad26.COV2.S vaccination with COVID-19-related hospitalization and death, adjusting for demographic characteristics, underlying health conditions, socioeconomic status proxies, and healthcare utilization. We found that by the end of 2022, 41% of surviving adults had completed vaccination and 8% had received a booster dose. Recent vaccination was associated with notable reductions in severe COVID-19 during periods dominated by Delta, and Omicron BA.1/2 and BA.4/5 (sub)lineages. During the latest Omicron BA.4/5 wave, within 3 months of vaccination or boosting, BNT162b2 and Ad26.COV2.S were each 84% effective against death (95% CIs: 57–94 and 49–95, respectively). However, distinct reductions of effectiveness occurred at longer times post completing or boosting vaccination. Results highlight the importance of continued emphasis on COVID-19 vaccination and boosting for those at high risk of severe COVID-19, even in settings with widespread infection-induced immunity.

## 1. Introduction

Globally, we have transitioned from emergency to long-term management of COVID-19 with reduced COVID-19 severity and high levels of population immunity from both prior infection and vaccination [1]. A key component of long-term management is optimizing vaccination strategies to protect those at high risk of severe COVID-19 as protection conferred by infection or vaccination against severe disease may wane [2,3,4,5,6,7] and there remains uncertainty about future SARS-CoV-2 evolution. However, the value of continued provision of vaccination opportunities is unclear in Africa and low- and middle-income countries (LMICs) where there were high levels of infection before vaccination became available [8]; achieving primary vaccine and especially booster uptake has been challenging [9], and there are competing health service priorities such as HIV and tuberculosis, many of which were adversely affected during the COVID-19 pandemic [10,11,12,13]. While most of those vaccinated in LMICs are likely to have hybrid immunity (prior infection and vaccination) [8,14,15], which has been shown to provide the most durable protection against severe disease in high-income country studies, there are few studies of long-term vaccine effectiveness in the context of very high levels of infection before vaccine availability, as experienced in countries such as South Africa (SA) [2]. While a health insurance scheme in SA showed rapid waning of protection of BNT162b2 vaccination against severe disease, with restoration by booster vaccination, applicability to the uninsured, poorer population is unclear [16]. Uninsured persons likely have higher levels of prior infection and a different comorbidity burden than the insured, and are more similar to populations from other LMICs. Real-world data on the durability of vaccine protection in these populations are needed to assess the benefit of ongoing COVID-19 vaccination in these contexts.

Our study aimed to estimate vaccine effectiveness, including by time since completing vaccination or boosting, against severe COVID-19 during each wave of COVID-19 disease, using observational data on a large general population in the Western Cape (WC) province of SA. The study also aimed to describe vaccine uptake. The Western Cape Department of Health and Wellness (WCDHW) regularly shared the latest results from this study with advisory groups to inform the provincial and national vaccination program. More specifically, for the estimation of vaccine effectiveness, by merging multiple data sources curated by the WCDHW, we studied the association of COVID-19 vaccination with three COVID-19 hospitalization and death outcomes amongst >2 million adult public sector healthcare users in the WC. We studied this association over time (from vaccine rollout through three subsequent COVID-19 waves through to the end of 2022) and differentiated vaccine state by the type of vaccine (Ad26.COV2.S or BNT162b2), completeness of primary vaccination, and time in the vaccine state.

## 2. Materials and Methods

### 2.1. Study Population, Study Period, and Data Sources

We studied adult (aged ≥ 18 years) public sector healthcare users (≥1 public sector healthcare encounter in the five years preceding 1 March 2020, i.e., the start of the COVID-19 pandemic) in the WC, who had a civil identification number enabling linkage to the national death registry. We excluded persons vaccinated before 17 May 2021 as earlier vaccination was restricted to healthcare workers.

The Western Cape Provincial Health Data Centre (WCPHDC) compiles de-identified person-level longitudinal data by integrating patient data from multiple information systems (administrative, laboratory, pharmacy, hospital, and disease management) in all public sector health facilities in the province [17]. The database, prior to any exclusions, is estimated to cover the three-quarters of the population of the WC who do not have private health insurance [18]. These data on public sector healthcare users are used to infer various health conditions with estimated onset dates. Additionally, for this analysis, the dates and types of administered COVID-19 vaccines were obtained from the national Electronic Vaccine Data System (EVDS) [19]. Data on all positive and negative SARS-CoV-2 PCR and antigen tests were obtained from public or private laboratories, or from consolidated laboratory data assembled by the National Institute for Communicable Diseases (NICD) [20]. WCPHDC public sector hospital data were supplemented by data on private sector COVID-19 admissions from the NICD hospital surveillance system (DATCOV) [21]. Data were linked to the national vital registration system to complete the ascertainment of mortality [22].

The dataset was released for analysis with all data linkage already conducted operationally by the data custodian (the WCPHDC). The data were released with a generated numeric patient identifier for each patient, which is unique to the dataset, preventing any cross-linkage with other data sources. Names and civil identifiers were not included. Given the importance of event timing, dates were not perturbed. The dataset was, however, only made available to the primary analyst under strict data management guidance. Basic data cleaning (such as the exclusion of persons with no reported sex or date of birth, or with dates lying outside of expected ranges) resulted in 0.65% of persons being excluded from the analysis; subsequently, 50.4% of persons who were indicated as not having a civil identification number (for linkage to the national death registry) were excluded.

Data were extracted during February 2023 after which SA has not experienced substantial surges in documented SARS-CoV-2 infections. The cohort was studied from 1 January 2020 until 31 December 2022 to allow for delays in outcome ascertainment. During this period, SA experienced five distinct COVID-19 waves with a different SARS-CoV-2 (sub)lineage dominating in each. For this study, each ‘wave’ period began when the dominant virus (sub)lineage first accounted for ≥20% of the week’s sequenced specimens nationally, and continued until the next wave. The five wave periods, derived in Appendix A correspond to the Ancestral (starting 1 March 2020), Beta (18 October 2020), Delta (30 May 2021), Omicron BA.1 or BA.2 (14 November 2021), and Omicron BA.4 or BA.5 (3 April 2022–2 July 2022) (sub)lineages.

The study was approved by the University of Cape Town Health Research Ethics Committee (HREC REF 460/2020) and WCGHW. Individual informed consent requirement was waived for this analysis of de-identified data.

### 2.2. Definition of Outcomes

We aimed to study severe COVID-19 and defined three COVID-19 outcomes, with each successive outcome (as ordered below) using a stricter definition of ‘severe’ disease, noting that some patients who died from COVID-19 were not hospitalized. Our outcomes were SARS-CoV-2 diagnoses associated with COVID-19-related (1) hospital admission (with disease of any severity), or death; (2) hospital admission with severe disease (i.e., requiring an intensive care unit or steroid prescription), or death; and (3) death alone. An admission or death was deemed COVID-19-related if sufficiently close to the SARS-CoV-2 diagnosis date with no record of non-natural causes—see definitions in Appendix A. SARS-CoV-2 testing was extremely limited in SA: seroprevalence studies suggest approximately 1 in 10 infections were diagnosed [14,15], and there was negligible use of self-tests during the study period. Therefore, ‘any SARS-CoV-2 diagnosis’ was not studied as an outcome.

### 2.3. Definition of Vaccine States

Vaccines became available to the general population on 17 May 2021, with the minimum eligibility age progressively reducing from 60 years to 12 years by 20 October 2021 [23]. For the primary vaccination series, persons either received the single-dose Janssen Ad26.COV2.S or two-dose Pfizer-BioNTech BNT162b2 vaccine, though immunocompromised individuals were eligible for an additional homologous vaccine dose. From the end of December 2021, adults were eligible for a booster vaccine from 60 and 180 (later reduced to 90) days after completion of the Ad26.COV2.S and BNT162b2 primary vaccination series, respectively, with opportunities for further boosting subsequently introduced.

In the analyses, persons could move through different vaccine states over time, from being ‘unvaccinated’ to briefly in a ‘transition’ state (<28 days after Ad26.COV2S or <21 days after BNT162b2) immediately after receiving their first dose of any vaccine, and then among three vaccinated states. At any time, a vaccinated person could have received a ‘Complete Ad26.COV2S’ (≥28 days after single dose), ‘Incomplete BNT162b2’ (no second dose, or <14 days thereafter), or ‘Complete BNT162b2’ (≥14 days after second dose) vaccination series. Among those with complete vaccination, we did not explicitly disaggregate by number of boosters due to limited booster uptake, and instead distinguished by ‘time in state’ (<3, 3–5, 6–8, and ≥9 months): this time started increasing from zero upon entering the vaccine state, and restarted (from zero) seven days after any booster dose. See Appendix A for detailed definitions. We considered only the type of the first vaccine, as very few persons received heterologous doses.

### 2.4. Statistical Methods

We described over time the rate of each COVID-19 outcome, as well as the proportions of those people with a documented SARS-CoV-2 infection who experienced each COVID-19 outcome, SARS-CoV-2 tests that were positive, and people in each vaccine state.

To estimate the association of vaccination with each outcome, we used a cox proportional hazards model with time-varying vaccine terms (vaccine state and time in state) and covariates, and parameterized it to allow the outcome rate to vary with calendar time. In SA, by definition, persons could not acquire a new SARS-CoV-2 infection within 90 days of the start of a previous documented infection, and were, therefore, removed from the at-risk population during these periods. People were censored at deaths that were not COVID-19-related. ‘Vaccine effectiveness’ (VE) was defined as one minus the model-estimated hazard ratio describing the rate of the outcome in a chosen vaccinated state versus in the reference unvaccinated state.

We adjusted for several covariates, and continuous covariates were categorised a priori. Demographic characteristics were sex and (time-varying) age. Time-varying health conditions were diabetes, hypertension, chronic kidney disease, chronic respiratory disease, HIV, ‘any’ tuberculosis (previous or ongoing), an ‘ongoing’ tuberculosis episode, pregnancy, and a prior SARS-CoV-2 diagnosis. Location (subdistrict/district) and location type for the most recent primary healthcare facility visit (within Cape Town Metro versus non-Metro or no recorded visit) were potential measures of socioeconomic status and healthcare access. The number (total for 5 years) and regularity (number of years with visits in 3 years) of primary healthcare facility visits before the pandemic (March 2020), and the number of negative SARS-CoV-2 tests more than 90 days before each analysis period (defined below) acted as proxies for other health risks and service use. All covariates were entered as main effects, though we used a likelihood ratio test (LRT) to investigate modification of VE by HIV status. Appendix A contains detailed definitions of covariates.

Given the complexity of the evolving virus, setting, and study population, we expected associations of COVID-19 outcomes with vaccination and covariates to change over our study period. We, therefore, fitted separate models to each of the analysis wave periods (defined above) from vaccine rollout, interpreting each estimated VE as an average VE over that period.

For supplementary analyses, we examined VE for distinct subpopulations based on sex, age, and HIV status at the start of 2020, and when excluding people who received heterologous doses. We also estimated VE over short 6-week rolling windows of time.

## 3. Results

### 3.1. Description of Cohort Characteristics, Vaccination, and COVID-19 Outcomes

Among 2,429,927 WC adult public sector healthcare users at the start of 2020, the median age of 38 years (quartile 1: 28, quartile 3: 52), and 60% were female (Table 1). There was a high burden of inferred non-communicable conditions (at the end of 2022, 23% and 10% had hypertension and diabetes, respectively; 8% and 4% had chronic respiratory disease and chronic kidney disease) and inferred infectious conditions (15% and 10% were living with HIV and had ever-experienced tuberculosis).

Only 40% of the cohort studied from 2020 was completely vaccinated by the end of 2022 (Table 2), and 8% received a booster. Heterologous doses were received by only 28 persons during primary vaccination, and by <1.5% of vaccinated persons during boosting. About 1 in 200 experienced COVID-19-related death; 1 in 150 experienced COVID-19-related hospitalization with severe disease, or death; and 1 in 60 experienced COVID-19-related hospitalization, or death. A total of 21% of COVID-19-related deaths occurred without a COVID-19-related hospitalization.

During the five COVID-19 waves of 2020–2022, weekly SARS-CoV-2 diagnoses reached a highest peak of 17 per 100 person years and test positivity reached 54%, with high overall mortality (6%) in those with documented SARS-CoV-2 infection (Figure 1A–C). Vaccines were available from mid-2021. By the start of 2022, 38% of the surviving cohort were completely vaccinated, with limited further vaccine uptake (Figure 1D). By the end of 2022, 5% of the surviving cohort had an incomplete vaccination series, and 41% had a complete series (71% BNT162b2), of whom only 3% completed or boosted vaccination within the last 6 months.

Comparing vaccination over time by age group (Figure 2), vaccine uptake was highest in the oldest persons (at the end of 2022, 33% and 51% were completely vaccinated among 18–34 and ≥60-year-old persons, respectively). During the Omicron periods (the end of 2021 through 2022), the completion of vaccination series and booster uptake were also highest in the oldest persons (at the end of 2022, 4% and 19% had boosters in 18–34 and ≥60-year-olds, respectively).

See Appendix A for tables of all analysis variables, including by age group, and see Appendix A for plots of outcomes, testing, and vaccination over time by sex, age group, and HIV status.

### 3.2. Vaccine Effectiveness Estimates

Before adjusting for time in a vaccinated state, there was substantial variation in estimated VE by COVID-19 outcome and wave period (Figure 3): outcome rates ranged from 20 to 92% lower in those completely vaccinated (versus unvaccinated). When accounting for time in a vaccine state, VE was high among those who most recently completed or boosted vaccination (i.e., within three months) for all wave periods and outcomes—point estimates were in the range of 54–92% for death, 51–92% for admission with severe disease or death, and 38–90% for any admission or death. However, there was a distinct reduction of VE with longer durations after completing or boosting vaccination—on average, VE point estimates reduced by about 25% and then 50% when moving from <3 months to 3–5 months, and then to 6–8 months post-vaccination, in turn, though uncertainties often became large at 6–8 months.

Considering the type of vaccine, within 3 months of completing or boosting vaccination in the Delta wave, at a point estimate level, the VE of BNT162b2 (90–92% across outcomes) was higher than for Ad26.COV2.S (62–67%), while the two vaccines’ VE were often more similar over the Omicron periods for similar times in a vaccine state. During the most recent Omicron BA.4/5 period and for the most severe outcome of death, within 3 months, BNT162b2 and Ad26.COV2.S were each 84% effective against death (95% CIs: 57–94 and 49–95, respectively).

We were unable to detect differences in VE by HIV status (*p*-values for LRTs by outcome and wave periods were mostly ≫ 0.05). Appendix A reports VE estimates for subpopulations, which all had reduced rates of outcomes when recently vaccinated, though uncertainties can be large. Appendix A presents estimated hazard ratios for all covariates in the multivariable cox models. Appendix A demonstrates the steady decline in VE measured over 6-week rolling windows of time that could be explained by factors such as increasing times in the vaccine state (not accounted for in this analysis), testing practices, and a growing mismatch between vaccine and dominant infection (sub)lineages. Appendix A presents a theoretical exploration of the magnitude of bias in estimating VE resulting from the under-ascertainment of prior infection.

## 4. Discussion

To our knowledge, this is the largest observational cohort study of COVID-19 VE in an African population with high levels of infection before vaccines became available and a low overall vaccination uptake; ongoing updates of this study’s results were used in real-time to inform SA’s vaccination programme. By the end of 2022, only 41% of adults had received primary vaccination and 8% a booster. VE against severe COVID-19 involving hospitalization or death was high within 3 months of completing or boosting vaccination, including during periods when more recent Omicron (sub)lineages dominated: BNT162b2 and Ad26.COV2S were >80% effective against death during the Omicron BA.4/5 period, and VE point estimates varied within 38–92% across distinct wave periods and outcomes. However, we found distinct reductions of VE at larger times post completing or boosting vaccination during the Omicron periods.

Our assessment of VE durability against severe COVID-19 following recent primary or booster vaccination in a setting of high levels of SARS-CoV-2 infection before vaccination provides an important extension of results from other settings: a systematic review including five studies of hybrid protection conferred by both confirmed prior infection and vaccination against severe COVID-19 (hospitalizations or severe disease) showed durable protective VE >90% (compared to persons unvaccinated without prior infection) for up to 12 months following primary vaccination and 6 months following boosting, before and during the Omicron period [2,24,25,26,27,28]. However, the review did not include any studies from Africa or settings with high levels of SARS-CoV-2 infection before Omicron emergence [5,25], follow-up after booster vaccination was limited, and results were variable [25,27].

Our results from a cohort study are not directly comparable with these mostly test-negative case-control studies, which examined the association of COVID-19 outcomes with confirmed hybrid protection (vaccination and prior infection) specifically in samples of only people with respiratory symptoms and testing for SARS-CoV-2 infection. Our study considered a general population, and, given the substantial under-ascertainment of SARS-CoV-2 infections in our setting, we could not robustly assess hybrid immunity. However, our results of high VE that reduced at longer durations post-vaccination were similar to three studies of severe COVID-19 during the Omicron period included in the review: In a Swedish study, VE from hybrid immunity was >80% after ≥three vaccine doses but 54% (95% CI: 13.75) with only two doses [28]. In a Czech study, VE conferred by hybrid immunity declined at >2 months since the last vaccine (primary series or booster) [25]. In a USA study, among people with prior infection and ≥5 months after primary vaccination, the VE of boosting was 56% (95% CI: 44.66) [29].

Strengths of our study include using routinely collected longitudinal observational data to assess real-world VE with a cohort design in a large population of relevance to many LMICs. We had robust ascertainment of vaccination through linkage to EVDS, and of SARS-CoV-2 testing, admissions, and deaths, through linkage to NICD records and vital registration. Our rich data also allowed us to adjust for known health conditions, health service utilisation, and testing behaviour.

Using routinely collected observational data also confers several limitations. First, there may be unmeasured confounding. For example, from a healthy vaccinee bias (inflating VE measures) [30] or greater health-seeking behaviour for COVID-19 disease among the vaccinated (deflating VE measures). Our adjustment for prior SARS-CoV-2 infections was also extremely limited—there was substantial under-ascertainment of infections in the WC throughout the pandemic, which likely increased with the emergence of the Omicron lineage [14]. Second, low booster uptake limited our ability to assess the effectiveness of boosters themselves or distinguish between homologous/heterologous boosting. Third, there may be a misclassification of outcomes. Identifying an outcome required a positive SARS-CoV-2 test in those admitted or deceased, which would be impacted by testing practices and the use of antigen (versus PCR) tests, which varied over time. We have previously shown a decrease in the specificity of identifying COVID-19-related hospitalizations and deaths with the emergence of Omicron BA.1/2 and widespread testing of all admitted patients [31,32]. This may contribute to a deflation in VE estimates in this study during the Omicron BA.1/2 wave compared to in the BA.4/5 wave when only those with clinically suspected COVID-19 were tested. Also, data on recorded steroid prescription to identify ‘admissions with severe disease’ were only available for public sector admissions. Fourth, we were only able to assess changes in VE with time post-vaccination during the later Omicron periods. Fifth, our study population of people who accessed public sector healthcare in the years preceding the pandemic and with South African civil identification numbers may not be representative of the remaining province’s uninsured persons who did not access healthcare over this period or who are foreign nationals. Lastly, this study, which was designed to estimate the association between vaccination and severe COVID-19 in our setting, was unable to explore vaccine safety in parallel. There are, however, extensive data from other settings that suggest that the benefits of COVID-19 vaccination with respect to COVID-19 disease, which align with those described in the current analysis, outweigh the risks of vaccination [33,34].

This study’s findings meaningfully add to the literature for informing policy decisions for COVID-19 vaccination programs. Since March 2024, adult vaccination in South Africa has ceased under the national program; any resumption within the public sector will depend on guidance from the National Advisory Group on Immunisation (NAGI) [35], access to appropriate vaccines, and the availability of funds [36]. This study’s findings support that sufficient consideration be given to the challenges of funding and access to vaccines in LMIC settings, and readiness to respond to any future COVID-19 waves, even in settings with widespread prior infection. Though of limited statistical power, the study also suggests that VE may be similar in the large population of people living with HIV in Southern Africa. As one of the few such studies from Africa and from LMICs overall during the Omicron period, findings suggest that a country such as South Africa could benefit from following the WHO SAGE roadmap [37] on COVID-19 vaccination, which recommends complete vaccination and regular boosting of vaccination in high-risk groups.

## 5. Conclusions

In our population with high levels of immunity from prior infection, recently completing or boosting COVID-19 vaccination (<6 months previously) provided protection against severe COVID-19, but this protection reduced at longer durations since the last vaccine dose. Continued opportunities for COVID-19 primary vaccination and boosting of vulnerable populations should be provided irrespective of levels of prior SARS-CoV-2 infection.

## Figures and Tables

**Figure 1 vaccines-12-00628-f001:**
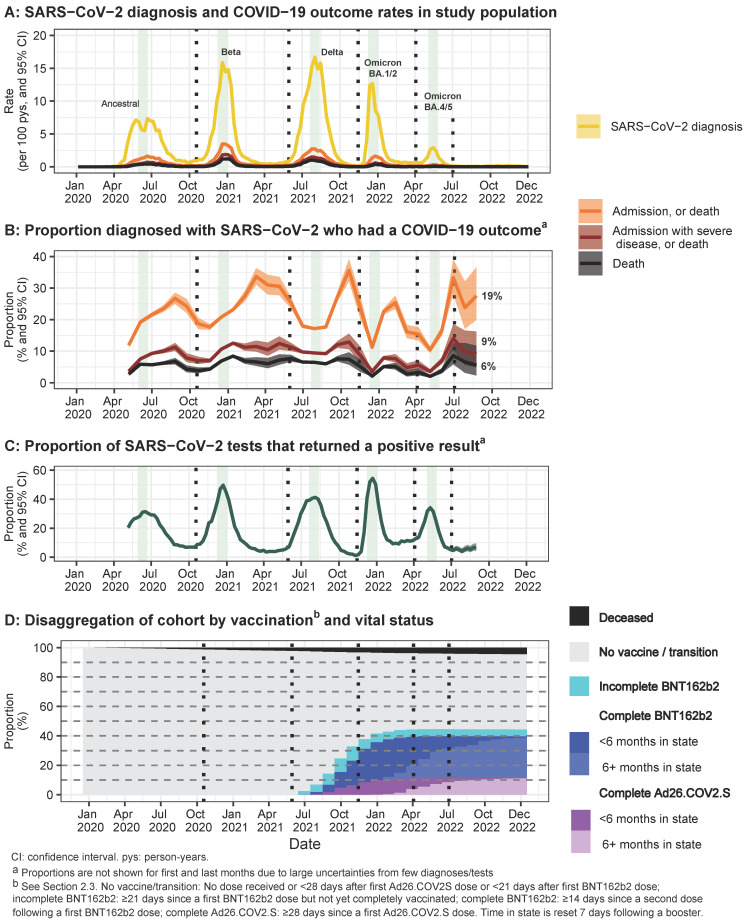
**COVID-19 outcomes, SARS-CoV-2 testing and positivity, and COVID-19 vaccination over time** in the cohort of South African adults utilising public sector healthcare in the Western Cape province: (**A**) Rates of SARS-CoV-2 diagnoses and COVID-19-associated hospitalization and death outcomes per week. (**B**) Proportions of persons with SARS-CoV-2 diagnoses who experienced COVID-19 outcomes per 4-week period, with overall proportions over the full plotted period indicated. (**C**) SARS-CoV-2 test positivity proportions per week. (**D**) Proportions of the population in different vaccine states at the end of each calendar month. Dashed vertical lines separate wave periods. Gray vertical bands indicate each wave’s peak SARS-CoV-2 diagnoses weeks.

**Figure 2 vaccines-12-00628-f002:**
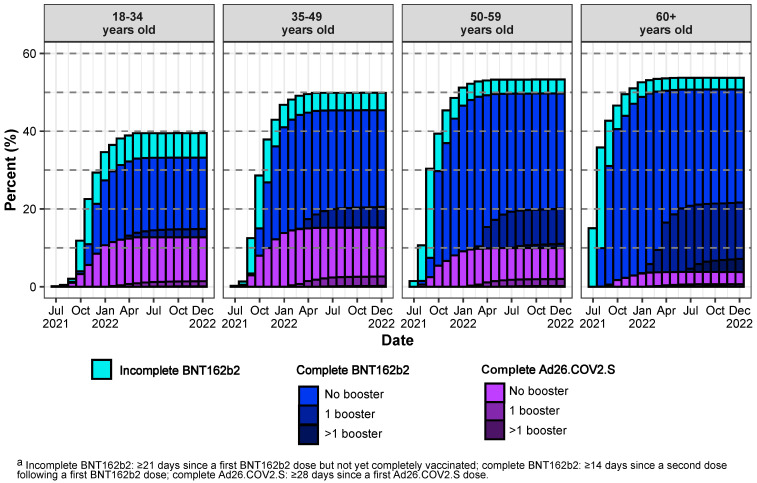
**Primary vaccination series and booster uptake** in the cohort of South African adults utilising public sector healthcare in the Western Cape province: proportions of the surviving population in the different vaccinated states ^a^ at the end of each calendar month (*x*-axis) by age group in years (based on age at the start of 2020) (figure columns).

**Figure 3 vaccines-12-00628-f003:**
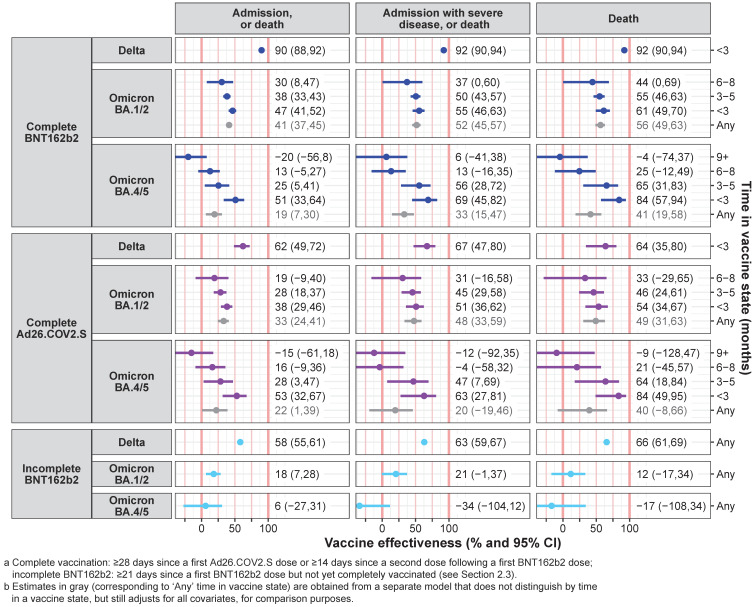
**Vaccine effectiveness estimates, by vaccine state, wave period, outcome, and time in vaccinated state** in the cohort of South African adults utilising public sector healthcare in the Western Cape province: Vaccine effectiveness estimates (*x*-axis) for different vaccinated states ^a^ and wave periods (figure rows), and COVID-19 outcomes (figure columns) by time ^b^ in the vaccinated state (*y*-axis). Vaccine effectiveness represents the reduction in outcome rates for vaccinated persons versus unvaccinated persons, adjusting for all covariates. Markers indicate point estimates. Horizontal error bars represent 95% confidence intervals.

**Table 1 vaccines-12-00628-t001:** Demographic characteristics and non-communicable and infectious conditions in the cohort of South African adults utilising public sector healthcare in the Western Cape province studied during calendar years 2020 through 2022, and stratified by completion of a primary COVID-19 vaccination series by the end of 2022. Percent (count frequency) is reported.

	Overall N = 2,429,927	Complete Vaccination by the End of 2022 ^a^	
Characteristics and Conditions (2020–2022)		No N = 1,455,385	Yes N = 974,542
**Age category at the start of 2020 (years)**			
18–34	41.6 (1,009,672)	46.6 (677,813)	34.1 (331,859)
35–49	29.3 (711,077)	27.2 (395,497)	32.4 (315,580)
50–59	14.1 (342,308)	12.4 (179,961)	16.7 (162,347)
60+	15.1 (366,870)	13.9 (202,114)	16.9 (164,756)
**Sex**			
Female	59.7 (1,450,660)	57.8 (841,578)	62.5 (609,082)
Male	40.3 (979,267)	42.2 (613,807)	37.5 (365,460)
**Hypertension**			
Absent at the end of 2022	76.5 (1,858,079)	79.8 (1,161,884)	71.4 (696,195)
Onset during 2020–2022	3.4 (83,174)	2.8 (40,473)	4.4 (42,701)
Present before 2020	20.1 (488,674)	17.4 (253,028)	24.2 (235,646)
**Diabetes**			
Absent at the end of 2022	89.5 (2,174,691)	90.8 (1,321,701)	87.5 (852,990)
Onset during 2020–2022	1.9 (46,111)	1.6 (22,634)	2.4 (23,477)
Present before 2020	8.6 (209,125)	7.6 (111,050)	10.1 (98,075)
**Chronic kidney disease**			
Absent at the end of 2022	95.7 (2,324,768)	95.9 (1,396,419)	95.3 (928,349)
Onset during 2020–2022	0.7 (16,445)	0.5 (7707)	0.9 (8738)
Present before 2020	3.7 (88,714)	3.5 (51,259)	3.8 (37,455)
**Chronic respiratory disease**			
Absent at the end of 2022	92.2 (2,241,094)	92.5 (1,346,451)	91.8 (894,643)
Onset during 2020–2022	1.5 (36,451)	1.4 (20,454)	1.6 (15,997)
Present before 2020	6.3 (152,382)	6.1 (88,480)	6.6 (63,902)
**HIV**			
Absent at the end of 2022	85.4 (2,075,796)	85.3 (1,241,925)	85.6 (833,871)
Onset during 2020–2022	1.4 (32,986)	1.4 (21,048)	1.2 (11,938)
Present before 2020	13.2 (321,145)	13.2 (192,412)	13.2 (128,733)
**Tuberculosis (at any time in past) ^b^**			
Absent at the end of 2022	90.2 (2,190,920)	89.2 (1,298,552)	91.6 (892,368)
(First) onset during 2020–2022	1.7 (42,068)	2.0 (28,783)	1.4 (13,285)
Present before 2020	8.1 (196,939)	8.8 (128,050)	7.1 (68,889)
**Tuberculosis (ongoing episode)**			
Absent throughout 2020–2022	96.8 (2,351,220)	96.2 (1,400,628)	97.5 (950,592)
Experienced an episode during 2020–2022	3.2 (78,707)	3.8 (54,757)	2.5 (23,950)

^a^ Complete vaccination: ≥28 days since a first Ad26.COV2.S dose or ≥14 days since a second dose following a first BNT162b2 dose (see Section 2.3). ^b^ Tuberculosis (at any time in the past) includes both ongoing and previous episodes.

**Table 2 vaccines-12-00628-t002:** COVID-19 vaccination, SARS-CoV-2 testing, and COVID-19 outcomes all as at the end of 2022 in the cohort of South African adults utilising public sector healthcare in the Western Cape province studied during calendar years 2020 through 2022, and stratified by completion of a primary COVID-19 vaccination series by the end of 2022. Percent (count frequency) is reported.

		Complete Vaccination by the End of 2022 ^a^	
State or Events Experienced as at the End of 2022	Overall N = 2,429,927	No N = 1,455,385	Yes N = 974,542
**COVID-19 vaccination status ^a^ and boosters**			
None	55.0 (1,336,826)	91.9 (1,336,826)	0.0 (0)
Incomplete BNT162b2	4.9 (118,559)	8.1 (118,559)	0.0 (0)
Complete Ad26.COV2.S (no booster)	9.7 (236,725)	0.0 (0)	24.3 (236,725)
Complete Ad26.COV2.S (and booster)	1.7 (41,196)	0.0 (0)	4.2 (41,196)
Complete BNT162b2 (no booster)	22.6 (548,914)	0.0 (0)	56.3 (548,914)
Complete BNT162b2 (and booster)	6.1 (147,707)	0.0 (0)	15.2 (147,707)
**SARS-CoV-2 diagnosis**			
Not experienced	92.1 (2,237,396)	93.3 (1,357,422)	90.3 (879,974)
Experienced	7.9 (192,531)	6.7 (97,963)	9.7 (94,568)
**Number of negative SARS-CoV-2 tests**			
0	81.8 (1,987,267)	84.3 (1,227,129)	78.0 (760,138)
1	13.1 (317,758)	11.4 (165,723)	15.6 (152,035)
2+	5.1 (124,902)	4.3 (62,533)	6.4 (62,369)
**COVID-19-related hospitalization, or death**			
Not experienced	98.4 (2,391,586)	98.2 (1,429,692)	98.7 (961,894)
Experienced	1.6 (38,341)	1.8 (25,693)	1.3 (12,648)
**COVID-19-related hospitalization with severe disease, or death**			
Not experienced	99.3 (2,412,279)	99.0 (1,441,138)	99.7 (971,141)
Experienced	0.7 (17,648)	1.0 (14,247)	0.3 (3401)
**COVID-19-related death and vital status ^b^**			
Alive	95.4 (2,318,859)	93.3 (1,357,339)	98.7 (961,520)
Died: not COVID-19-related	4.1 (99,525)	6.0 (87,001)	1.3 (12,524)
Died: COVID-19-related	0.5 (11,543)	0.8 (11,045)	0.1 (498)

^a^ Complete vaccination: ≥28 days since a first Ad26.COV2.S dose or ≥14 days since a second dose following a first BNT162b2 dose; incomplete BNT162b2: ≥21 days since a first BNT162b2 dose but not yet completely vaccinated (see Section 2.3). ^b^ Based on the current (i.e., time-varying) complete primary vaccination status of the person (rather than at the end of the analysis period): the non-COVID death rate per 100 person years in unvaccinated and vaccinated persons was 1.4 and 1.0, respectively; the COVID death rate in unvaccinated and vaccinated persons was 0.2 and 0.04, respectively.

## Data Availability

The data are not publicly available due to privacy and ethical restrictions. Requests to access the datasets should be directed to Western Cape Provincial Health Data Centre.

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
