# Peer review of "COVID-19 Vaccine Uptake and Effectiveness by Time since Vaccination in the Western Cape Province, South Africa: An Observational Cohort Study during 2020–2022"

_vaccines, 2024, doi:10.3390/vaccines12060628_

Round 1
Reviewer 1 Report
Comments and Suggestions for Authors
I have reviewed with interest the manuscript titled: "COVID-19 vaccine uptake and effectiveness by time since vaccination in the Western Cape province, South Africa: An observational cohort study during 2020-2022."
The introduction is correct; however, we recommend clarifying the study's objectives more precisely and concretely.
The methodology needs to explicitly outline the mechanisms of data anonymization, especially given the extensive nature of the database.
It would be insightful to know the authors' estimation of how many patients may have been lost from the registry due to not being registered in the civil identification database used for death registration.
It is advisable to clearly inform the reader what percentage of the population corresponds to those analyzed in their database. The graphs are accurate and reader-friendly, and the statistical analysis is appropriate. We miss the study's limitations being clearly stated.
The results are predictable, not very new. It would be advisable for the authors to convey the practical application of their analysis for the new post-pandemic era we are living in.
Author Response
Thank you for the time you have taken to review our article. Please see the attachment.

Reviewer 2 Report
Comments and Suggestions for Authors
The proposed manuscript is interesting from the point of view of the current information about the effectiveness of vaccination and data on the incidence of coronavirus infection. This study is distinguished by the breadth of parameters and approaches studied, including different waves of strains and comorbidities (as evidenced by additional information).
There are several questions concerning the study.
1. Lines 139. “all adults could receive additional booster doses 180 days after last doses.” Why is such a period selected, is it a local indication or is there a rule on re -vaccination after 180 days? Please explain in a pair of sentences.
2. Do the authors have any data on the effectiveness of vaccination or data on morbidity/mortality in patients with autoimmune diseases? It would be good to include this because almost no one provides such data. At this stage, as a wish from the reviewer. Traditionally, data on diabetes and kidney failure are included, but the percentage of autoimmune diseases (arthritis, multiple sclerosis) is higher among different segments of the population.
3. Line 348/ Conclusion. Can the authors conclude that the use of vaccination is higher than the risk of consequences? Can vaccination be used as a regular part of the vaccination schedule, or can it be used only during periods of epidemic?
Author Response
Thank you for taking the time to review our article. Please see the attachment.
